# B Cells versus T Cells in the Tumor Microenvironment of Malignant Lymphomas. Are the Lymphocytes Playing the Roles of Muhammad Ali versus George Foreman in Zaire 1974?

**DOI:** 10.3390/jcm9113412

**Published:** 2020-10-24

**Authors:** Minodora Desmirean, Sebastian Rauch, Ancuta Jurj, Sergiu Pasca, Sabina Iluta, Patric Teodorescu, Cristian Berce, Alina-Andreea Zimta, Cristina Turcas, Adrian-Bogdan Tigu, Cristian Moldovan, Irene Paris, Jakob Steinheber, Cedric Richlitzki, Catalin Constantinescu, Olafur Eysteinn Sigurjonsson, Delia Dima, Bobe Petrushev, Ciprian Tomuleasa

**Affiliations:** 1Department of Hematology, Iuliu Hatieganu University of Medicine and Pharmacy, 400124 Cluj Napoca, Romania; dr.minodora.desmirean@gmail.com (M.D.); rauchsebastian@yahoo.com (S.R.); ancajurj15@gmail.com (A.J.); pasca.sergiu123@gmail.com (S.P.); iluta.sabina@yahoo.com (S.I.); patric_te@yahoo.com (P.T.); Burancristinam@gmail.com (C.T.); jakob.steinheber@googlemail.com (J.S.); cedric-richlitzki@web.de (C.R.); constantinescu.catalin@ymail.com (C.C.); 2Department of Pathology, Constantin Papilian Military Hospital, 400124 Cluj Napoca, Romania; ireneparis2002@yahoo.com; 3Medfuture Research Center for Advanced Medicine, Iuliu Hatieganu University of Medicine and Pharmacy, 400124 Cluj Napoca, Romania; cristi.berce@gmail.com (C.B.); zimta.alina.andreea@gmail.com (A.-A.Z.); tiguadrianbogdan@yahoo.com (A.-B.T.); moldovan.cristian1994@gmail.com (C.M.); bobe.petrushev@gmail.com (B.P.); 4Department of Anesthesia and Intensive Care, Iuliu Hatieganu University of Medicine and Pharmacy, 400124 Cluj Napoca, Romania; 5The Blood Bank, Landspitali—The National University Hospital of Iceland, 101 Reykjavik, Iceland; oes@landspitali.is; 6School of Science and Engineering, Reykjavik University, 101 Reykjavik, Iceland; 7Department of Hematology, Ion Chiricuta Clinical Cancer Center, 400124 Cluj Napoca, Romania; deli_dima@yahoo.com; 8Department of Pathology, Octavian Fodor Regional Institute of Gastroenterology and Hepatology, 400124 Cluj Napoca, Romania

**Keywords:** malignant lymphomas, tumor microenvironment, lymphocyte inter-talk, B lymphocytes, T lymphocytes

## Abstract

Malignant lymphomas are a heterogeneous group of malignancies that develop both in nodal and extranodal sites. The different tissues involved and the highly variable clinicopathological characteristics are linked to the association between the lymphoid neoplastic cells and the tissues they infiltrate. The immune system has developed mechanisms to protect the normal tissue from malignant growth. In this review, we aim to explain how T lymphocyte-driven control is linked to tumor development and describe the tumor-suppressive components of the resistant framework. This manuscript brings forward a new insight with regard to intercellular and intracellular signaling, the immune microenvironment, the impact of therapy, and its predictive implications. A better understanding of the key components of the lymphoma environment is important to properly assess the role of both B and T lymphocytes, as well as their interplay, just as two legendary boxers face each other in a heavyweight title final, as was the case of Ali versus Foreman.

## 1. Background on the Tumor Microenvironment in Malignant Lymphomas

In malignant lymphomas, there is a constant fight between lymphoma-promoting and anti-lymphoma immune cells, which ultimately results in the abnormal supremacy of pro-malignant cells which are seen and recognized by diagnostic pathologists each time a lymph node biopsy is sent for interpretation. This constant fight is mirrored in the boxing match of Muhammad Ali versus George Foreman in Zaire 1974, when after a long game of equal powers in the third round of the match, unexpectedly, Forman with a huge punch sent Ali flying into the ropes. It was obvious to all 60,000 fans in the Stade du 20 Mai in Kinshasa, Zaire, and millions more watching live on television, that things were beginning to go horribly wrong for Muhammad Ali. Foreman pursued Muhammad Ali around the ring, launching punches from odd angles and power shots that thudded into Ali’s body, inhuman blows that seemed too much for any man to withstand. However, in the end, Ali recovered and won the game through a final knockout.

The same struggle is also seen when analyzing the slides of a lymphoma case, since the differential diagnosis between B-cell lymphoma, T-cell lymphoma, or Hodgkin’s lymphoma is probably one of the most challenging analysis that a pathologist has to make, since it is based only on hematoxylin-eosin staining and the evaluation of cell morphology. Pathologists find it difficult to know whether a B lymphocyte or a T lymphocyte is malignant or an accompaniment cell, in the absence of any staining. As follows, many times 10–12 different staining protocols are carried out, until the final diagnosis is reached.

Lymphoid tumors are malignant proliferations of B, T, or natural killer (NK) cells. They often have a variable clinical behavior. Some are indolent cell proliferations, whereas others have an aggressive behavior and a fulminant evolution [1,2,3,4,5]. T-cell lymphomas, in most cases, are classified into the second category, being aggressive tumors with increased resistance to treatment and more frequent and faster relapse rates when compared to B-cell lymphomas [6]. In comparison to tumors originating from B lymphocytes, for which therapy in the last years became more precise with a better response and remissions lasting longer [7,8,9,10,11], the therapy in T-cell lymphomas (TCL) generally consists of a combination of classical chemotherapy with cyclophosphamide, doxorubicin, vincristine and prednisone (CHOP) protocol combination chemotherapy).

As stated in the beginning, the tumor microenvironment in lymphomas is built on a constant fight between bystander cells. As follows, in the past years, research did not just focus on tumor cells, but also on the tumor microenvironment (TME). The TME sustains and promotes cancer cell development and progression and it is involved in resistance to chemotherapy. Liu et al. [12] have proven that tumor cells from a Hodgkin lymphoma (HL), also known as a Reed-Sternberg or Hodgkin cell, collaborate closely with the surrounding lymphocytes, mastocytes, and other cells from the TME, with the aim of secreting molecules necessary for cellular survival and division [12]. In some indolent lymphomas, like mucosa-associated lymphoid tissue (MALT) lymphomas, chronic inflammation induced by bacterial infections in most of the cases, as is the case of infections by *Helicobacter pylori*, contributes to the promotion and support of tumor development. According to the van den Berg group, from the University of Groningen, based on their experience, this hypothesis was proven by the response to antibiotic treatment, which does not just lead to the eradication of the infection, but also to a cease of tumoral development. The TME is of crucial importance in lymphoma development, as it is constantly sustained by its heterogenous composition and stimulated by malignant cells [13]. Thus, the TME is dependent on intercellular signaling between tumoral cells and surrounding cells, capable of changing their phenotype as a result of lymphoma cells acquiring new genetic mutations. The TME is formed by a multitude of cells, which include tumor-associated macrophages, tumor-associated fibroblasts, follicular dendritic cells or dendritic cells, as well as immune cells, as is the case of cytotoxic T lymphocytes (CTLs), T follicular helper cells (TFH), regulatory T cells (Tregs), natural killer (NK) cells, and bystander B lymphocytes (Figure 1). Silencing the host’s immune system is an important feature of malignant lymphomas. Achieving a better understanding of distinct pathways of interactions between lymphomas and different immunological microenvironment compounds yields substantial potential for new treatment concepts. In both B-cell and T-cell lymphomas, tumor cells as well as their infiltrating immune cells upregulate several immune checkpoint genes and critical proteins in a distinct pattern of several immune escape strategies. Although, overactivation of NF-κB and B-cell receptor (BCR) represents major cell intrinsic determinants of lymphoma aggressiveness, by mediating immune escape. The co-expression of programmed death (PD-1) and programmed death ligand (PD-L1) contributes further to giving B-cell lymphomas the worst prognosis of the lymphoma subtypes. These molecules are important tools to control T-cell activity and proliferation and can both inhibit T cells as well as stimulate immunosuppressive regulatory T cells, as further presented in this work.

## 2. Immune Cells of the Tumor Environment

One of the most important cell subtypes in the lymphoma microenvironment is regulatory T cells (Tregs). Wang et al. [14] have shown that in the lymphoma microenvironment, Tregs can be classified into direct tumor-killing Tregs, suppressor Tregs (CD8-positive), incompetent Tregs, and malignant Tregs (FOXP3-positive) [14]. Besides the immunosuppressive effect on T lymphocytes, Tregs may also cause the suppression of B lymphocytes, macrophages, dendritic cells, or even NK cells [6]. Non-Hodgkin’s lymphoma (NHL) frequently has a high number of Tregs [15]. Thus, the subtype of predominant Tregs and the rate of survival in these patients may be linked. On the other side, in follicular lymphomas, a strong correlation between the survival rate and the number of intrafollicular Tregs (FOXP3-positive) was reported with a high number of intrafollicular Tregs correlated with an unfavorable prognosis and survival, as according to Xie et al. [16].

Cytotoxic T lymphocytes (CTLs) are a subtype of CD8-positive T lymphocytes with a crucial role in infectious and malignant diseases. They are activated by a diversity of stimuli, may it be infectious agents or tumoral antigens, including most MHC antigens [17]. Most CTLs express surface cytotoxic markers, out of which T-cell intracellular antigen (TIA) and TIA receptor (TIAR) both have a role in promoting stability or suppressing translation of messenger RNA, involved in apoptotic and inflammatory processes. The downregulation of TIA and TIAR leads to uncontrollable cell growth through incompletely understood mechanisms. Wahlin et al. [18] reported the association between an increased number of CD8-positive lymphocytes and better survival in follicular lymphomas. Carreras et al. [19] supported this and also showed that an increased number of cytotoxic programmed cell death protein 1 (PD-1) lymphocytes is associated with a better prognostic factor in case of follicular lymphomas [19].

The cytotoxic activity of T lymphocytes is potentiated through activating the PD-1 pathway, which leads to the apoptosis of tumoral cells [20]. In oncogenesis, the activation of a protein called lymphocyte activation gene (LAG-3) leads to the suppression of T-lymphocyte activation and therefore secretion of cytokines, by which the immune homeostasis is maintained. The mechanism of signaling and interaction through which LAG-3 works with checkpoints is still uncertain. LAG-3 exerts an inhibitory effect on most lymphocytes, a synergy with PD-1, and therefore, an inhibitory anti-tumoral immune response. Once the T lymphocytes during tumor progression have been exhausted, this phenomenon is accompanied by the increment of inhibitory receptors for LAG-3. The interaction between LAG-3/ major histocompatibility complex (MHC II) inhibits the expansion of T lymphocytes and suppresses cytokinetic response. Similarly, LAG-3 could potentiate the sensibility of tumor cells to Tregs in case of a clinical relapse through mediating the inhibition of T lymphocytes. LAG-3 is present in normal human physiology in small quantities, in inactive CD8-positive lymphocytes, although their level can rise significantly when responding to a tumoral antigen [21]. The blockage of LAG-3 on CD8-positive T lymphocytes leads to the enhancement of their function and implies an increased production of interferon (IFN-γ), which portrays a phenomenon independent on T lymphocytes [22,23,24]. Furthermore, CD8-positive T lymphocytes can express LAG-3 concomitantly with other co-inhibitory immune checkpoints, as is the case of PD-1 [25].

Follicular helper T (TFH) cells are a population of T lymphocytes. These cells were first isolated from the palatine tonsils, hence playing a role in immunity by sustaining B lymphocytes in the production of antibodies. TFH cells are present in tissues rich in B lymphocytes, especially in the germinal center (GC). TFH cells have a crucial part in the interaction with B lymphocytes, starting from the maturation up until the stimulation of those for expressing immunomodulatory effects. Likewise, TFH cells play a role in the formation of the GC by mediating interactions between CD40 and the CD40 ligands on B lymphocytes, through the production of IL-21 and thus leading to the proliferation of B lymphocytes. TFH cells express surface markers, amid CXCR5, PD-1, inducible co-stimulatory protein (ICOS), and CD200. These markers are used in the diagnosis of angioimmunoblastic T-cell lymphoma (AITL), a lymphoma originating from TFH cells. However, pathology slides of such lymphomas also include plasmocytes, immature B lymphocytes, chemokines, immunoglobulins, IL-21, and IL-6, thus originating in the largest part by TFH tumors [26,27,28]. Likewise, peripheral T-cell lymphomas, not otherwise specified (PCTL-NOS) and the follicular PCTL variant are lymphomas with TFH phenotypic T cells [29]. AITL is thus a prototype of TFH lymphomas. These lymphoma subtypes were described in the 1970s as a non-neoplastic, immunological condition, and were later recognized as a subtype of the peripheral T-lymphoma. Recent genetic studies have demonstrated the fact that changes in AITL genotype are the reason for different genetic mutations, without the rest of the T-cell lymphomas, especially peripheral T-cell lymphoma (PTCL) [30,31]. The follicular variant of PTCL was recently described, and the name comes from the follicular architecture of the lesions that resemble a follicular, lymphoid structure mimicking a similar architecture like B-cell follicular lymphomas. Considering the phenotype of TFH, similar common clinical and pathological traits between AITL and the follicular variant of PTCL have been reported [32]. In support of this theory, genetic data prove the presence of the chromosomal translocation t(5;9)(q33;q22), present in the majority of the follicular variant of PTCL cases, but absent in the majority of non-follicular PTCLs, including AITL [33].

The activation of T lymphocytes is based on the antigen-presenting cell (APC) capacity to internalize and present a molecular MHC II antigen. Dendritic cells (DC) have similar roles to APCs and certain B lymphocytes, thus presenting a specific B-cell receptor (BCR) able to interact with T-lymphocyte-specific antigens. As BCRs are produced by a variety of B lymphocytes, most B cells play a minor role in presenting T-lymphocytic antigens. Some cells like T lymphocytes, NK cells, and B lymphocytes can transfer cell membranous and cellular components to others. The most important difference between B lymphocytes is their antigen-specific BCR, through which a change of the membrane would allow the transfer of antigen-presenting capacity through the transfer of BCR. Through the transfer of BCR from antigen-specific B lymphocytes to bystander B lymphocytes during the immune response, the capacity of B lymphocytes to connect and present antigens can be potentiated, displaying a process amplifying the immune response mediated by T lymphocytes [34,35].

## 3. T Cell and B Cell Subpopulation Types and Their Interplay in Lymphoma

The interplay between T cells and B cells in lymphoma is highly dependent on the type of malignancy, meaning that, the anti-tumoral effect of cytotoxic T or B cells is reversed dependent on lymphoma origin. For instance, the CD8+ (cytotoxic) T cells suppress the progression of various lymphomas, such as Epstein-Barr virus (EBV)-positive Hodgkin’s lymphoma [36], follicular lymphoma [18], and B-cell non-Hodgkin’s lymphoma [37]; however, in CD8+ lymphomas, these cells are malignantly transformed and their proliferation/stimulation is associated with disease progression [38,39]. Moreover, the regulatory T cells, that are considered pro-tumoral factors of local immunity [40], have a limited anti-tumoral effect in cytotoxic T-cell lymphoma because it can specifically interact and target the malignant cells [40]. As opposed to solid cancer types, the Tregs in lymphoma can have three roles: negative, neutral, or positive. For instance, high FOXP3+ Treg presence in T-cell lymphoma has no effect on the overall survival rate of the patients [41].

In fact, according to Wang et al. [14], there are multiple types of regulatory T cells [14]. The generally encountered suppressor Tregs decrease patients’ survival rate, by downregulating the local immune response at the malignant lymphoma site, in diseases such as non-Hodgkin’s lymphoma, peripheral T-cell lymphoma, anaplastic large cell lymphoma, and Hodgkin’s lymphoma [14]. The second type of Tregs with pro-tumoral effects are the Tregs that have undergone malignant transformation and are now the source of T-cell lymphoma [14]. Thirdly, there is a small subset of Tregs that slow down tumor progression. These are the direct tumor-killing Tregs that in T-cell lymphomas inhibit directly the proliferation and aggressiveness of T cells [14,42]. Lastly, the fourth type of Tregs are the incompetent Tregs that have lost the capacity to exert any effect in the local immunity interplay [14,43].

The T-helper cells have a dual role depending on their type. The T-helper cells of type 1 (Th1) activate specifically the cytotoxic T cells (CD8+) through antigen presentation; thus, they have an anti-tumoral effect, especially in classical Hodgkin’s lymphoma, B-cell non-Hodgkin’s lymphoma (NHL) [44], and complete remission of diffuse large B-cell lymphoma [45]. T-helper 2 cells (Th2) are differentiated T lymphocytes important in the immune response against pathogens that do not directly infect cells. These cells also play key roles in tissue repair and contribute to the pathophysiology of allergic disorders.

The Th2 cells on the other hand disrupt the activity of Th1 cells and thus, their abundance is associated with a lower disease-free survival in B-cell lymphoma [44], especially in untreated B-cell lymphoma [45]. The Th17 cells have a dual role in lymphomas. These cells have an anti-tumoral effect because they generally inhibit the activity of Tregs. This has been proven especially in EBV-negative classical Hodgkin’s lymphoma (CHL) [36]. This T-cell subtype induces B-cell differentiation to IgG2a and IgG3 subtype, general B-cell proliferation, and higher production of novel antibodies [46]. In classical B-cell lymphoma, circulating Th17 cells have a lower number in diagnosed patients versus healthy individuals. However, in B-cell lymphoma patients [47], it was proven that the Th17 cells sustain rituximab resistance [48], having a higher number in recurrent disease [47]. This may be caused by the fact that Th17 cells stimulate B-cell proliferation independent of B-cell normal or lymphomagenesis status.

The B-cell population also establishes multiple interactions with normal or malignant T cells. The B2 follicular B cells interact with memory helper T cells and cause their anergy, thus allowing the progression of malignant B cells in mature B-cell lymphoma [49]. IgA-positive B cells function as antigen-presenting cells for T cells and activate CD8-positive T cells, thus functioning as anti-tumoral stimulators in lymphomas [50]. In concomitant lymphoplasmacytic lymphoma and plasma cell myeloma, the IgA-producing plasma cells are the origin of malignancy, thus they have a pro-tumoral effect in this case [51].

The memory B (IgG-positive/IgM-positive) cells are localized in the tumor where they secrete tumor-specific antibodies that activate innate immunity. The IgG-positive B cells can produce granzyme B and directly inhibit the progression of malignant cells or through their secretion of IFN-γ, cooperate with CD8-positive T cells to directly kill malignant cells [50]. However, in case of B-cell lymphoma protein (BCL2): IGH translocation, the IgG-positive B cells re-enter in the germinal center and develop into malignant follicular lymphoma [52]. In EBV-positive Burkitt’s lymphoma, memory B cells help the EBV infection to stay latent [53]. The IgM-positive memory B cells are early memory B cells that go through malignant transformation over the instalment period of mantle-cell lymphoma [54]. The regulatory B cells are a subpopulation of B cells that specifically secrete large quantities of IL-10 that causes downregulation of local immune response [55,56]. These cells have been found in large numbers in non-Hodgkin’s lymphoma [57]. These cells have a pro-tumoral effect by suppressing the activity of cytotoxic T cells and Th1 cells [55]. Details on the role of these subpopulations of T and B cells are depicted in Table 1 [58,59,60,61,62,63].

## 4. Therapeutic Targets for T or B Cell Subpopulations in Lymphoma

The presented lymphocyte subpopulations establish multiple interactions with malignant cells. This results in either suppression of immune response in immunogenic T or B cells, or further stimulation of the activity of immunosuppressive cells. Programmed cell death protein 1 (PD-1) [66] and cytotoxic T lymphocyte-associated protein 4 (CTLA-4) are two well-known receptors present on the surface of T-helper cells, especially Th1, and produce their anergy. Basically, these cells are no longer capable of exerting their anti-tumoral response. The anti PD-1/PD-L1 or CTLA-4 have significant effects on preventing Th1 anergy and reactivating the effects of cytotoxic T cells. The therapy with anti-PD-L1 significantly improves the progression-free survival of lymphoma patients that have PD-L1 overexpression on the surface of malignant cells [66]. In phase I or phase II clinical trials, PD-L1 blockage was especially efficient in relapsed/refractory B-cell non-Hodgkin’s lymphomas or in primary mediastinal large B-cell lymphoma. Moreover, in follicular T-cell lymphoma, because the malignant cells exhibit a higher level of PD-L1, this therapy also has significant good results [67].

The CTLA-4 is the other T-cell receptor that causes T-cell anergy and that is also overexpressed on Tregs [68,69]. In Hodgkin’s lymphoma, this receptor is more expressed than PD-1 in recurrent disease [70]. What may be difficult for future targeting of this receptor is its presence in the intracellular vesicles of FOXP3+ Tregs, where it cannot interact with conventional therapeutic antibodies [71]. As follows, when considering a more targeted therapy toward preventing T-cell anergy, other components of synaptic T cell–cancer cell interactions should be considered, as these may be alternative pathways that give therapeutic resistance and lymphoma relapse. Adenosine A2a receptor (A2aR) is another receptor present on cytotoxic T cells and T-helper cells that downregulate the T-cell response and act in synergy with CTLA-4 and PD-1 [72]. Patients with diffuse large B-cell lymphoma have a very high expression of this receptor [73]. Lymphocyte-activation gene 3 (LAG-3) is present on CD8+ T cells and T-helper cells [74]. Its interaction with major histocompatibility complex type II (MHC II) causes CD8+ T-cell exhaustion in synchrony with PD-1 receptor in B-cell non-Hodgkin’s lymphoma [74,75], extranodal NK/T-cell lymphoma [76], Hodgkin’s lymphoma (HL) [77], and malignant Hodgkin Reed-Sternberg (HRS) [77]. LAG-3 acts as an off-switch in T cells. In immune active T-helper cells, it downregulates their reactivity, while in Tregs, it prevents their immunosuppressive response [74,75].

T-cell immunoglobulin and mucin domain-containing protein 3 (TIM3) acts in a similar manner and many times in coordination with PD-1 and CTLA-4. This receptor is expressed in IFN-γ-producing T-helper cells and cytotoxic T cells. Its interaction with interleukin-inducible T-cell kinase (ITK) downregulates IL-2 production and causes T-cell anergy [78,79]. In T-cell lymphoma, TIM3 overexpression does not affect patients’ survival rate, but it stimulates acquired chemoresistance [80]. In follicular B-cell non-Hodgkin’s lymphoma, it is associated with lower survival rate and advanced disease stage [64].

There are contradictory data on V-set domain-containing T-cell activation inhibitor 1 (VTCN1) that is present on a minor population of T cells, being mostly specific for B cells. However, some reported the absence of this receptor on B or T cells or its presence only in stimulated lymphocytes [81]. This receptor, together with PD-1 and CTLA-4 activation, significantly lowers the secretion of IFN-γ, T-cell proliferation, and cytotoxic response [82]. It was found to be overexpressed in lymphoma cells from non-Hodgkin’s patients [83]. B7-H3 is highly expressed in IFN-γ-producing T cells. This biomarker is overexpressed in many malignancies, being related to metastasis, epithelial-to-mesenchymal transition, and disease progression [84]. However, through its stimulation of cytotoxic T cells, it can stimulate local immunity in vivo [85]. In extranodal nasal natural killer (NK)/T-cell lymphoma (ENKTCL) [86] and mantle cell lymphoma [87], this marker is also overexpressed and has been associated with disease progression and acquired chemoresistance. The B7-H3-reactive chimeric antigen receptor T cell (CAR T) cells or the anti-B7-H3/CD3 bispecific antibodies showed high efficiency in vivo studies. B- and T-lymphocyte attenuator (BTLA) is expressed on the surface of follicular T cells, Th1, and B cells [88]. This receptor binds to herpes virus entry mediator (HVEM) and downregulates CD8+ T cells, thus preventing their cytotoxic activity. In diffuse large B-cell lymphoma with high expression of BTLA, this checkpoint inhibitor co-expresses with PD-1, TIM3, and LAG-3. When activated, BTLA prevents CD8+ T-cell differentiation [89]. Its overexpression is also associated with poor prognosis for patients with T-cell lymphoma [90]. In B-cell lymphoma from germinal center, loss-of-function mutations in HVEM and the disrupted interaction between this receptor and BTLA stimulates the proliferation of malignant B cells; thus, in this case, BTLA is a tumor suppressor [91].

In cutaneous T-cell lymphoma, diffuse large B-cell lymphoma, mantle cell lymphoma, B-cell non-Hodgkin’s lymphoma (NHL) [92], follicular lymphoma [93], T-cell immunoreceptor with Ig and immunoreceptor tyrosine-based inhibition motif (ITIM) domains (TIGIT) is overexpressed. This receptor is present on T-cell population of various types: Th1, TFH, CD8+ T cells, and Tregs. In Tregs, TIGIT activation stimulates the immunosuppressive phenotype of Tregs; in the immunoactive T-cell population, it causes cell exhaustion, sometimes in synergy with PD-1 receptor [74]. V-domain Ig suppressor of T-cell activation (VISTA) is a receptor present in the whole population of tumor-infiltrating lymphocytes; however, its activation suppresses specifically only the T-cell response, while stimulating the expression of FOXP3. The activation of this checkpoint inhibitor has no effect over B-cell response. This biomarker lowers the survival rate of lymphoma patients [94]. The blockage of VISTA and PD-1 through the use of a small molecule, called CA-170, increases T-cell activation and IFN-γ production by T cells [95]. Still, in Krakow, Musielak et al. [96] have shown that CA-170 has no interaction with PD-1/PD-L1 and state that this previously potential clinical application must be further investigated [96].

More details on the role of the abovementioned biomarkers, viewed as therapeutic targets, as well as the currently available immune checkpoint inhibitors, and development and testing stages are presented in Table 2 [96,97,98,99,100,101,102,103,104].

## 5. Extracellular Components in the Lymphoma Microenvironment

Angiogenesis has a crucial part in tumor progression and survival [105,106]. Malignant cells are capable of synthesizing stimulatory factors of angiogenesis. Newly formed vessels have a structural defect and differ from normal vessels, that are more immature and permeable, which enables tumor cells to penetrate the vascular walls and enter the bloodstream, thereby increasing the risk of metastasis [107,108,109,110]. To get the required rate for cellular multiplication, lymphoma cells need a new vascular network dedicated to satisfying the energy needs of the cells [111]. The production of immature tumor vessels is caused by an abnormal secretion of vascular growth factors by the tumor cells: vascular endothelial growth factor (VEGF), platelet-derived growth factor (PDGF-β), and transforming growth factor (TGF-β) [112]. A defect model of the vascular network from the TME leads to cell hypoxia and decreased supply of the inflammatory cells in the surrounding of tumors and indirectly of chemotherapeutics, but also increases tumor cell adaptability, rendering them more therapy-resistant and aggressive [108]. New therapies focus more on correcting structural and functional defects of the network of neoangiogenesis [113]. Thus, by repairing the vascular defects, the supply of oxygen normalizes, preventing hereby the adoption of more aggressive cellular components caused by a hypoxic TME [114].

The extracellular matrix (ECM) is a component of the TME and a complex system of macromolecules that guarantees a biochemical fundament and the biology necessary for cellular survival. Tumor cells have the key feature of plasticity, which allows them to change their physiological characteristics for surviving a therapy [115]. Tumor cells are also capable of producing, degrading, or modifying molecular components of the ECM, with the goal to facilitate tumor progression [116]. In solid tumors, alterations of the ECM accelerate tumor progression, tumor invasion, and metastasis, either by cellular modification at the level of tumor cells or by effecting the TME [117,118].

Cytokines are a broad and heterogeneous class of bioactive molecules that are secreted by cells of the immune system and influence a target cell [119]. We describe the role of cytokines in one type of PTCL, namely angioimmunoblastic T-cell lymphoma (AITL), because of the heterogeneous aspect of both cytokines and peripheral T-cell lymphoma (PTCL). AITL is associated with autoimmune events induced by this disease and because of its development from a T follicular helper cell (TFH), which has a high involvement in cytokine signaling [120]. In a murine model of the disease, AITL was shown to be driven by proteins specific for TFH cells [121]. A key cytokine characteristic to TFH cells and in turn to AITL is IL-21 [122]. IL-21 is upregulated in AITL and its knockdown has been shown to inhibit lymphoma genesis in a model of Swiss Jim Lambart (SJL) mouse [123]. TFH cells also secrete IL-10, linked to a negative prognosis in AITL due to higher levels of M2 macrophages. [124]. The cytokines secreted by AITL cells also have a role in chemoattraction and activation of immune cells infiltrating AITL. For B cells, the upregulation of the CXCL13/CXCR5 axis recruits B cells in the AITL microenvironment [31,125]. Other cells known to infiltrate AITL are CD8-positive T cells and Th17 cells, the latter being recurrently implicated in autoimmune events and being stimulated by IL-6 produced by mast cells in the microenvironment [126,127]. By directly synthesizing IL-6, mast cells contribute to setting a pro-inflammatory TME. The AITL lymphoma clone itself attracts mast cells in this inflammatory microenvironment and thus, mast cells change the immunological TME of AITL, with important clinical implications in diagnosis and prognosis.

Recently, international attention was gained in cancer biology by the study of extracellular vesicles in a wide range of biological fluids [65,128,129,130,131,132,133,134]. These vesicles are heterogeneous particles that originate from different subcellular compartments and have different molecular dimensions and composition. According to their origin, extracellular vesicles are classified into microvesicles (which are formed via outward budding of the plasma membrane), exosomes (which are formed in multicellular bodies through the intraluminal budding of endosomal membranes), and apoptotic bodies (specific to cell death process). Extracellular vesicles are important mediators of cell–cell communication and play a key role in both normal and pathological processes, such as cancer development and progression [135].

Exosomes are nanovesicles secreted by normal or tumor cells with dimensions between 30–150 nm [136,137,138,139,140]. During biosynthesis, exosomes carry important bioactive molecules from the donor cell, such as nucleic acids (DNA, RNA, microRNA, long non-coding RNA (lncRNA), circulating RNA), proteins, and lipids [141]. The exosomal membrane is formed from cholesterol, sphingolipids, and phosphatidylserines—molecules found on the outward cell membrane and on the outside of the exosomes. The proteins of the exosomal membrane are tetraspanins (CD9, CD63, and CD81), major histocompatibility complex molecules, or proteins involved in the cell adhesion process [142].

In contrast, large amounts of cytoskeletal and heat shock proteins, as well as endosomal proteins (SNARE, annexin, and flotillin), Alix, and TSG101, which belong to the biosynthesis of multicellular bodies [143], are encapsulated in the exosome structure. Nucleic acids are encapsulated in the exosome structure and further involved in the mediation of important biological processes that underlie the development and progression of cancer cells, as well as in processes of angiogenesis, migration, and invasion, immune system, and metastatic lesion formation.

Exosomes derived from tumor cells can influence normal cells to “create” a tumor microenvironment that allows tumor growth and metastasis. Thus, tumor exosomes influence endothelial cells to support neoangiogenesis, a process that fuels tumor growth and induces vascular permeability, in order to facilitate metastasis [144]. Exosomes are also key players in the process of differentiating fibroblasts by transforming them into pro-angiogenic and pro-tumorigenic cancer-associated fibroblasts [145,146]. By interfering the exosomes with the immune system, the target cell phenotype is altered to pro-tumorigenic [147] and pre-metastatic [144]. By transferring oncoproteins to a target cell, exosomes induce changes in phenotype by activating different signaling pathways, such as mitogen-activated protein kinase (MAPK) and PI3K-AKT-mTOR. Exosomes can transfer oncogenic entities, such as mutated proteins [148], fusion genes (EML4-ALK) [149], and oncogenic long non-coding RNA [150,151] from cancer cells to the tumor microenvironment cells. By transferring microRNA sequences, exosomes induce resistance to receptor cell level therapy [150] and induce hypoxia [152] and angiogenic response in endothelial cells.

The lymphoma microenvironment is important in lymphoma biology as it promotes tumor cell proliferation and resistance to apoptosis and provides the mechanisms to prevent the immune system. The lymphoma microenvironment is formed from immune cells, stromal cells, cytokines, blood vessels, and extracellular matrix components (Figure 2). Moreover, extracellular vesicles form a two-way flow of information between the microenvironment and the nanoparticles [135]. Recently, it was shown that extracellular vesicles secreted by a diffuse large B-cell lymphoma (DLBCL) contain the mutated form of myeloid differentiation primary response 88 (MYD88), a part of reprogramming the bone marrow lymphoid microenvironment [153]. Up to date, most studies have assessed the role of extracellular vesicles (EVs) secreted by lymphomas on immune cells. For EBV-induced lymphomas, EVs polarize the resident macrophages to an M2 phenotype, which leads itself to an immune-evasive microenvironment [154]. Moreover, the same EVs also induce apoptosis in T cells, further increasing the immune-evasive phenotype of the lymphoma [155]. Hodgkin’s lymphoma (HL) cells secrete CD30-positive EVs, which interact with cells presenting CD30 ligand (CD30L), ultimately leading to changes in the immune cells [156]. Some lymphomas also secrete EVs that contain NKG2D ligand, which leads to a downregulation of NKG2D on NK cells in the lymphoma microenvironment, thus inhibiting NK-mediated tumor killing [157]. Aside their roles in inhibiting immune cells already present in the lymphoma microenvironment, lymphoma-secreted EVs also upregulate endothelin B on endothelial cells, leading to an inhibition of T-cell migration [158,159].

## 6. Conclusions

On 30 October 1974, a historic boxing event in Kinshasa, Zaire (now called Democratic Republic of the Congo), was organized at the 20 May Stadium. It brought face-to-face the undefeated world heavyweight champion George Foreman against challenger Muhammad Ali, the former heavyweight champion. The event had an attendance of 60,000 people. Ali won by knockout, putting Foreman down just before the end of the eighth round. It has been called “arguably the greatest sporting event of the 20th century”, watched by a record estimated television audience of 1 billion viewers worldwide. Without crossing the line, in tumor biology, very few cases present the background of two similar players, such as B and T lymphocytes interacting with each other in malignant lymphomas. The tumor microenvironment and the local immune system of the niche have been merely regarded as the background for neoplastic cells to become a focus of our understanding of cancer development, cancer progression, and point of action for new therapeutic concepts. Still, the interplay between B cells and T cells play a role in the pathogenesis of various lymphoma subtypes. The plasticity of the microenvironment makes it difficult to be studied in static circumstances and, for obvious reasons, in animal models as the constitution of the microenvironment and its interactions might differ from that in humans. Still, progress has been achieved for the contribution of immune system to lymphoma development and progression, and this knowledge has been transferred into therapeutic strategies. In the present work, we presented the key cellular players and their role in the development, dissemination, and response or resistance to therapy.

## Figures and Tables

**Figure 1 jcm-09-03412-f001:**
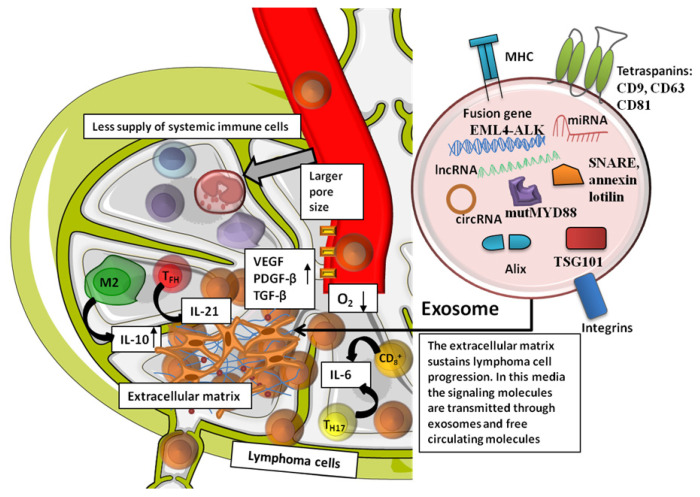
The role of extracellular matrix in lymphoma development and progression, an exemplification of the sequence of events happening inside the malignant lymphoid tissue. The vascular network is disorganized, and it has larger pores compared to normal blood vessels, due to overexposure to vascular endothelial growth factor (VEGF), platelet-derived growth factor (PDGF-β), and transforming growth factor (TGF-β). Due to these characteristics, the local microenvironment is hypoxic and there is a general lack of systemic immune cell provision, while the local immune cells secrete cytokines meant to facilitate lymphoma cell proliferation and invasion. The lymphoma cells (represented in the current figure in round brown) are at a higher number than all the other non-malignant immune cells in the lymphoma microenvironment. They arrive in the lymphoid organ through extravasation from general blood circulation, helped by locally modulated immunity, and the lymphoma cells proliferate and become more aggressive, until they start to leave the lymph node and enter into the lymphatic circulation. Inside the malignant lymphoid tissue, the macrophages are polarized to M2 phenotype and secrete interleukin-10 (IL-10), meaning that they have anti-inflammatory and malignancy-promoting properties. The T follicular helper (TFH) cells secrete IL-21, and the T helper (Th) Th17 and CD8+ T cells secrete IL-6. Th17 cells have a dual role by inhibiting regulatory T cells (Tregs), but also stimulating malignant proliferation of B cells. CD8+ cells (cytotoxic T cells, CTL) inhibit lymphoma progression, but they are in a small number in the lymphoma microenvironment. However, the main mechanism through which the lymphoma progression is sustained by the extracellular environment is due to the secretion of exosomes by the immune cells (T or B lymphocytes) and the lymphoma cells (of B or T origin). In the top right part of the figure, in the circle, there is a representation of an exosome.

**Figure 2 jcm-09-03412-f002:**
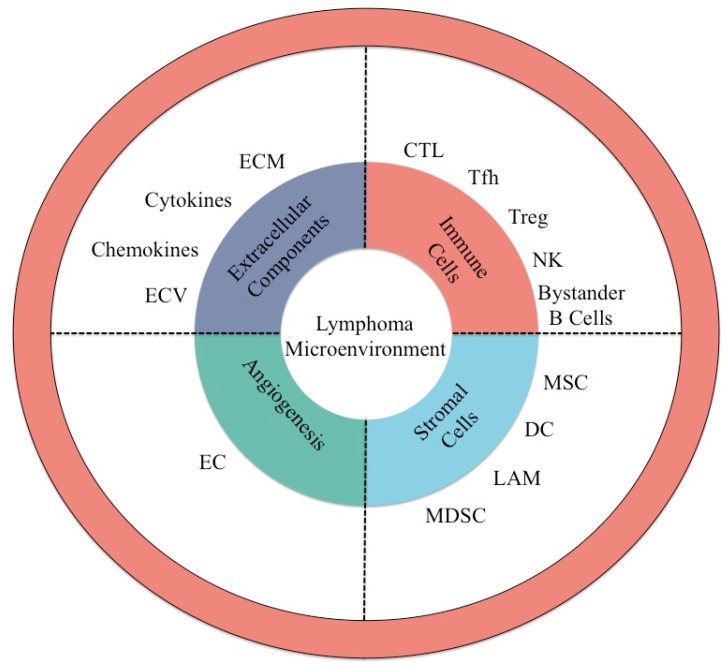
The role of extracellular matrix in lymphoma development and progression. The vascular network is disorganized and it has larger pores than normal blood vessels, due to overexposure to VEGF, PDGF-β, and TGF-β. Due to these characteristics, the local microenvironment is hypoxic and there is a general lack of systemic immune cell provision, while the local immune cells secrete cytokines meant to facilitate lymphoma cell proliferation and invasion. The macrophage is polarized to the M2 phenotype and secretes IL-10, the TFH cells secrete IL-21, and the Th17 and CD8+ T cells secrete IL-6. However, the main mechanism through which the lymphoma progression is sustained by the extracellular environment is due to the secretion of exosomes by immune cells (T or B lymphocytes) and lymphoma cells (of B or T origin). The exosomes contain at their surface, targeting molecules, such as tetraspanins (CD9, CD63, and CD81), major histocompatibility complex (MHC) (I and II), and integrins. In their interior, the exosomes have nucleic acids: DNA in the form of fusion genes (EML4-ALK), long non-coding RNAs (lncRNAs), circular RNAs (circRNAs), and miRNAs. The exosomes also contain SNARE, annexin, lotilin, Alix, TSG101, and mutated forms of proteins, such as mutMYD88.

**Table 1 jcm-09-03412-t001:** Pro-/anti-tumoral immune cells in the tumor microenvironment of T-cell or B-cell lymphoma.

Cell Type	Activity	Target	Type of Lymphoma	Reference
Suppressor Treg	Pro-tumoral	Suppression of CD8+ T cells	Non-Hodgkin’s lymphoma, peripheral T-cell lymphoma, anaplastic large cell lymphoma, Hodgkin’s lymphoma	[14]
Malignant Treg	Pro-tumoral	Inhibition of CD8+ activity	T-cell lymphoma	[14]
Direct tumor-killing Treg	Anti-tumoral	Are a source of malignant cells	Cutaneous T-cell lymphoma, follicular lymphoma, diffuse T-cell lymphoma, extranodal NK/T-cell lymphoma	[14,42]
Incompetent Treg	Anti-tumoral	No effect	Angioimmunoblastic T-cell lymphoma	[14,43]
Cytotoxic T cell	Anti-tumoral, Pro-tumoral in CD8+ lymphomas	Inhibition of malignant cells; inhibited by Tregs	Epstein-Barr virus (EBV)-positive Hodgkin’s lymphoma, CD8+ cytotoxic T-cell lymphoma, nodal cytotoxic T-cell lymphoma, Hodgkin’s lymphoma, follicular lymphoma, B-cell non-Hodgkin’s lymphoma	[36,37,38,39,58,64]
Follicular helper T cell	Anti-tumoral, Pro-tumoral in follicular T-cell lymphomas	B-cell maturation	Peripheral T-cell lymphoma, angioimmunoblastic T-cell lymphoma (AITL), nodular lymphocyte predominant Hodgkin’s lymphoma	[59,60,65]
T-helper cell type 1 (Th1)	Anti-tumoral	Activation of CD8+ T cells	EBV-negative classical Hodgkin’s lymphoma (CHL), abundant in classical Hodgkin’s lymphoma, B-cell non-Hodgkin’s lymphoma (NHL), abundant in complete remission of diffuse B-cell large cell lymphoma	[36,44,45]
T-helper cell type 2	Pro-tumoral	Inhibition of Th1	Depleted in classical Hodgkin’s lymphoma, B-cell non-Hodgkin’s lymphoma (NHL), abundant in untreated B-cell diffuse large cell lymphoma	[44,45]
T-helper cell type 17	Anti-tumoral (mostly), pro-tumoral through rituximab resistance	Contradictory conversion with Tregs, stimulates B-cell proliferation and antibody production	EBV-negative classical Hodgkin’s lymphoma (CHL), B-cell lymphoma, rituximab-resistant B-cell lymphoma	[36,46,47,48,61]
B2 follicular B cells	Pro-tumoral	Memory CD4(+) T cells	Probably present in mature B-cell lymphoma	[49]
Immunoglobulin A (IgA)+ plasma cell	Pro-tumoral	Activation of CD8+ T cells, antigen-presenting cells	Concomitant lymphoplasmacytic lymphoma and plasma cell myeloma, peripheral T-cell lymphoma associated with IgG plasma cell leukemia and IgA hypergammaglobulinemia	[50,51,62]
IgG1+/IgM memory B cells	Anti-tumoral general, pro-tumoral in case of malignant accumulation	Co-operation with CD8+ T cells	Malignant follicular lymphoma (oncogenic role of memory B cells with BCL2: immunoglobulin heavy chain (IgHV) translocation), Burkitt’s lymphoma (EBV+), mantle cell lymphoma	[50,52,63]
Regulatory B cells	Pro-tumoral	Suppression of cytotoxic T cells and Th1 cells; activation of malignant T cells	Non-Hodgkin’s lymphoma	[55,57]

**Table 2 jcm-09-03412-t002:** Types of molecular targets that are currently used or have the potential of being used as therapeutic targets of immune checkpoint inhibitors.

Marker	Cell	Drug	Effect	Clinical Effect in Lymphoma Patients	Status	Reference
A2AR	CD4+ and CD8+ T cells, Tregs in response to stress	Monoclonal antibodies (mAbs): SCH58261, SYN115, ZM241365, and FSPTP	It downregulates T-helper cells and cytotoxic T cell response, synergy with cytotoxic T-lymphocyte associated 4 (CTLA-4) and PD-1	Lower survival rate in diffuse large B-cell lymphoma patients with high expression of A2AR	Phase I	[72,73]
B7-H3	IFN-γ-producing T cells	Potential therapeutic benefits of monoclonal antibodies (mAbs) targeting this biomarker	It increases T-cell reactivity	Increased chemoresistance and tumor progression of mantle cell lymphoma	Preclinical data	[85,87,96]
BTLA	T follicular helper cell differentiation, Th1 helper cell	Soluble herpes virus entry mediator (HVEM) ectodomain protein (solHVEM) through CART cells or bispecific antibody delivery could restore HVEM– B and T lymphocyte attenuator (BTLA), resulting in apoptosis and tumor growth delay in B-cell lymphomas	It binds to the HVEM receptor, downregulates CD8+ T-cell cytotoxicity in diffuse large B-cell lymphoma, suppresses minor histocompatibility antigen-specific CD8+ T cell, downregulates B-cell response	Poor overall survival in patients with T cells highly expressing BTLA Highly expressed in small lymphocytic lymphoma HVEM loss-of-function mutations frequently occur in B-cell lymphomas from germinal center (GC), leading to disruption of HVEM–BTLA interaction, and stimulation B-cell proliferation Upregulated in advanced stages of peripheral T-cell lymphomas (PTCLs)	Preclinical results	[88,89,90,97]
CTLA-4	Intracellular vesicles in FOXP3+ Treg cells or activated T cells	Ipilimumab, tremelimumab	It prevents conventional T-cell activation, it is highly expressed on Tregs and stimulates their activity	Hodgkin’s lymphoma has higher CTLA-4+ T cells than PD-1+ T cells and it is associated with recurrent disease Super-expressed in relapsed/refractory B-cell lymphoma	Phase I clinical trial	[68,69,70,98]
LAG-3	Activated CD8+ T cell, T-helper cells	mAbs: BMS-986016, LAG525, MK-4280, and IMP321 (APC activator)	Its overactivation in tumor microenvironment causes T-cell exhaustion, especially in collaboration with PD-1 activation	Highly expressed in Hodgkin’s lymphoma (HL) and malignant Hodgkin Reed-Sternberg (HRS) cells Poor disease-free survival in lymphoma patients High presence in extranodal NK/T-cell lymphoma	Phase I-II clinical trial in solid tumors, only preclinical data for lymphoma	[74,75,76,77,99]
PD-1	CD4+ T cell	Nivolumab, pembrolizumab	It causes T-cell anergy	Improved overall and progression-free survival, especially in PD-L1+ lymphomas	Phase I and II clinical trial	[66,100]
TIGIT	Activated CD4+, CD8+ T cells, T follicular helper (TFH) cells, Tregs	mAbs: MTIG7192A, BMS-986207, OMP-313M32, MK-7684, AB154, CGEN-15137, and CASC-TIGIT	TIGIT+ Tregs are more potent, CD8+ T cells have a much weaker response, PD-1 and TIGIT have a synergic effect	Increased in the sera of cutaneous T-cell Lymphoma patients Expressed in the tumor microenvironment in diffuse large B-cell lymphoma patients, and to a smaller proportion of cases in mantle cell lymphoma Overexpressed in follicular lymphoma tissue Might be the reason for PD-1 inhibition failure in B-cell non-Hodgkin’s lymphoma (NHL)	Phase I in solid tumors	[92,93,99,101,102]
TIM3	IFN-γ-producing CD4+ and CD8+ T cells, Tregs	mAbs: Sym023, INCAGN02390, LY3321367 ± LY3300054, Sym021 ± Sym023, MBG453 ± PDR001, BGB-A425 + tislelizumab, TSR-022 ± TSR-042, TSR-022 + TSR-042 + chemo; bispecific mAbs: RO7121661, LY3415244	Its overexpression on T lymphocytes is specific for CD8+ T-cell exhaustion. Single expression shows weak exhaustion, while co-expression with PD-1 has a more pronounced effect	Associated with higher histological grade (grade 2–3), greater lactate dehydrogenase (LDH) level in the blood, lower overall survival rate in follicular B-cell non-Hodgkin’s lymphoma	Phase I-II clinical trial in solid tumors, only Sym023 is in Phase I clinical trial for lymphoma	[99,103,104]
VISTA	Highly expressed on tumor-infiltrating leukocytes	CA-170 V-domain Ig suppressor of T cell activation (VISTA) proteins	Suppressed proliferation of T cells, but not B cells	Possible poor survival rate of lymphoma patients	Phase I and II clinical trial	[94,96]
VTCN1 (B7-H4/B7- S1)	Activated minor population of T cells, most B cells (defined as B220+), macrophages	Potential therapeutic benefits of mAbs targeting this biomarker	It decreases IFN-γ production by T cells, downregulates cytotoxic T-cell response, decreases T-cell proliferation, synergy with PD-1 and CTLA-4	Preclinical results: improved survival rate in B-cell lymphoma of mice, smaller tumor size of mice T-cell lymphoma Overexpressed in non-Hodgkin’s lymphoma cells isolated from patients	Preclinical studies in solid tumors	[81,82,83]

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
