# Peer review of "B Cells versus T Cells in the Tumor Microenvironment of Malignant Lymphomas. Are the Lymphocytes Playing the Roles of Muhammad Ali versus George Foreman in Zaire 1974?"

_jcm, 2020, doi:10.3390/jcm9113412_

Round 1

Reviewer 1 Report

The review article well describes microenvironment of of malignant lymphoma tissue with infiltrating lymphocytes. Tumor-infiltrating lymphocytes (TIL) has been well discussed from the aspect of tumor immunity. In recent years, cytotoxic T cells are focused via the clinical usage of immune check point inhibitors. This review manuscript would be helpful for understanding the biology of tumor immunity. 

I did not point out the major concerns. 

A few minor issues I was aware of...

  1. This review looks to be written by multiple authors. Then some technical terms are inconsistent. For instances, T follicular helper cells (TFH), or follicular helper T cells (TFH), which is the one? 
  2. Typo is there, for instances... Page 5, line 6 "CD8-pozitive" should be substitute to "CD8-positive," Page 15, line 27 "proliferative T-cell lymphoma" should be revised to "peripheral T-cell lymphoma."
  3. The format of the text is inconsistent. 
  4. Terminology is not international standard. For instances, programmed cell death protein 1 (PD-1), cytotoxic T-lymphocyte associated antigen 4 (CTLA-4)...

Author Response

Reviewer 1:

The review article well describes microenvironment of malignant lymphoma tissue with infiltrating lymphocytes. Tumor-infiltrating lymphocytes (TIL) has been well discussed from the aspect of tumor immunity. In recent years, cytotoxic T cells are focused via the clinical usage of immune check point inhibitors. This review manuscript would be helpful for understanding the biology of tumor immunity.

I did not point out the major concerns.

Thank you for the appreciation of our manuscript.

A few minor issues I was aware of...

This review looks to be written by multiple authors. Then some technical terms are inconsistent. For instances, T follicular helper cells (TFH), or follicular helper T cells (TFH), which is the one?

Thank you. It is follicular helper T cells. We have corrected the manuscript.

Typo is there, for instances... Page 5, line 6 "CD8-pozitive" should be substitute to "CD8-positive," Page 15, line 27 "proliferative T-cell lymphoma" should be revised to "peripheral T-cell lymphoma."

Thank you. We have corrected the manuscript.

The format of the text is inconsistent.
Terminology is not international standard. For instances, programmed cell death protein 1 (PD-1), cytotoxic T-lymphocyte associated antigen 4 (CTLA-4).

Thank you. We have corrected the manuscript.

Reviewer 2 Report

This is a very comprehensive review article by Desmirean and colleagues with an aim to better explain the B versus T cells role in the tumor microenvironment of malignant lymphomas. However, I have few concerns.

Although I appreciate the effort for being innovative and creative, I am not sure that the reference to the historic boxing event is appropriate (in the title, introduction, and conclusion section).

The manuscript needs major language editing.  I would strongly encourage the authors to use formal medical English rather than colloquial expressions, for example: ''When pathologists look at a slide, it is awfully hard to determine…''

Moreover, there are many repetitions of the same word, for instance: ''Lymphoid tumours are malignant proliferations of B, T or NK cells, with a variable genotype and phenotype. They present with a variable behaviour…''

''Background on the tumor microenvironment in malignant lymphomas'' section is hard to follow. Perhaps due to its somewhat narrative style. In general, more persuasive, and argumentative approach is needed. In addition, the authors do not mention immune escape strategies of lymphomas. I think it is important to include it in the background section and elaborate it later in the manuscript.

Table 1 and Table 2 are very informative and practical.

Author Response

Reviewer 2:

Comments and Suggestions for Authors
This is a very comprehensive review article by Desmirean and colleagues with an aim to better explain the B versus T cells role in the tumor microenvironment of malignant lymphomas. However, I have few concerns.

Although I appreciate the effort for being innovative and creative, I am not sure that the reference to the historic boxing event is appropriate (in the title, introduction, and conclusion section).

Thank you for the appreciation of our manuscript. Still, we would very much rather to keep the analogies with the boxing event, as it illustrates very good the interplay between B and T cells. The manuscript thus has a new, catchy title, introduction, and conclusion, that will be read and cited by physicians and scientists. We deeply appreciate your understanding and support.

The manuscript needs major language editing.  I would strongly encourage the authors to use formal medical English rather than colloquial expressions, for example: ''When pathologists look at a slide, it is awfully hard to determine…''
Thank you. We have corrected the manuscript.

Moreover, there are many repetitions of the same word, for instance: ''Lymphoid tumours are malignant proliferations of B, T or NK cells, with a variable genotype and phenotype. They present with a variable behaviour…''
Thank you. We have corrected the manuscript.

''Background on the tumor microenvironment in malignant lymphomas'' section is hard to follow. Perhaps due to its somewhat narrative style. In general, more persuasive, and argumentative approach is needed. In addition, the authors do not mention immune escape strategies of lymphomas. I think it is important to include it in the background section and elaborate it later in the manuscript.

Thank you. We have corrected the manuscript and added that “Silencing the host’s immune system is an important feature of malignant lymphomas. Achieving a better understanding of distinct pathways of interactions between lymphomas and different immunological microenvironment compounds yields substantial potential for new treatment concepts. In both B-cell and T-cell lymphomas, tumor cells as well as their infiltrating immune cells upregulate several immune checkpoint genes and critical proteins in a distinct pattern of several immune escape strategies. Although over-activation of NF-κB and B-cell receptor (BCR) represent major cell-intrinsic determinants of the lymphoma aggressiveness, by mediating immune escape, the co-expression of PD-1 and PD-L1 contributes further to giving B-cell lymphomas the worst prognosis of the lymphoma subtypes. PD1/PDL1 is the best studied and most frequently therapeutically used pathway of immune evasion. These molecules are important tools to control T-cell activity and proliferation and can both inhibit T-cells as well as stimulate immunosuppressive regulatory T-cells, as further presented in the manuscript.“. And we further discuss the role of PD-1 and PD-L1 in the chapter “Therapeutic targets for T or B cells subpopulations in lymphoma.”

Table 1 and Table 2 are very informative and practical.

Thank you for the appreciation of our manuscript, including Tables 1 and 2.

Reviewer 3 Report

The review focused on discussion microenvironment of lymphomas and interaction of different component of niche and immune cells during lymphoma formation and homeostasis.  The review will be interesting for scientists from a broad field of tumour research including, haematologists, immunologists, cellular biologists, vascular biologists and clinical doctors. The manuscript is well written and well-structured It has a good flow of text. This review presents some interesting data but there are obvious limitations and I suggest some points, which could be considered by the authors for manuscript revision.

Major comments.

  1. “This hypothesis was proven by the response to antibiotic treatment, which does not just lead to the eradication of the infection but also to a cease of tumoral development” This strong statement is not finish by paper citation or comment like “it is author opinion based on their data”.
  2. “In contrary to B-cell lymphomas and Hodgkin’s lymphomas (HL), TCL show an increased heterogeneity regarding the TME”. Not clear in which regards TCL heterogeneity is increased in comparison with TME
  3. Piece of the text below is not clear and have to be explanted by separation on “logical blocks”. “Independent from the subtype of TCL, the TME is considered as an entity of stimulation, sustenance, and promotion of lymphomas. Thus, by having a heterogenous composition, is a complex system that is dependent on the stimulation from tumor cells (13). The whole TME is dependent on the cell signals received by the tumour cells, which are also able to change their function because of genetic mutations”.
  4. Sentence “These are cytotoxic T lymphocytes (CTLs), T follicular helper cells (TFH), regulatory T cells (Tregs), natural killer (NK) cells and bystander B-Lymphocytes, as shown in Figure 1.” Author miss until this point the important component of TME. e.g tumor associated macrophages (TAMs) stromal components like fibroblasts, follicular dendritic cells, dendritic cells.
  1. On Figure 1 subtitle not described well. Most of component of the picture not indicated (e.g. vessels, M2-macrophages, Th17 lymphocytes, CD8+ -> CTL etc.) It is not clear all detail of the Figure. What big round cells mean. I guess this discription of lymphoid tissue (like lymph node). it is not discribed in subtitels or text of the manuscript.
  2. “Recent data shows that in some lymphoma subtypes, cancer cells act similar to Tregs”. Similar to what? Please explain and supply reference.
  3. “ Tregs are a subpopulation of CD4-positive T lymphocytes …”. The authors claim that Treg is CD4 + and describe them directly below as CD8 +.
  4. Phrase is not clear «…. TME is most extensively studied in non-Hodgkin’s lymphomas (NHL), where the tumour cells mostly present as reduced from the tumoral mass».
  5. Statement: “On the other side, in follicular lymphomas a strong correlation between the survival rate and the number of intrafollicular Tregs (FoxP3-positive) was reported with a high number of intrafollicular Tregs correlated with an unfavourable prognosis and survival.” has no reference present.
  6. Authors use the term CD4 + as a synonym for T-regulatory cells, while the authors describe also T-helper cells that have the same phenotype (CD4+). Authors should be more precise in determining the phenotype of cells. Alternatively, they can remove CD4 as an exclusive definition of T-regulatory cells.
  7. Authors describe “The T helper cells of type 1 (Th1) activate specifically the cytotoxic T cells (CD8+) through antigen presentation..” however they not describe what Th2 cells role in immunity and directly mention only “The Th2 cells on the other hand disrupt the activity of Th1” it is better, at least shortly, describe role of Th2.
  8. Authors cited that “The blockage of VISTA and PD-1 through the use of small molecule, called CA-170, increases the T cell activation and IFN-γ production by T cells” however it was recently shown that CA-170 has no interaction with  PD-1/PD-L1 binding https://pubmed.ncbi.nlm.nih.gov/31374878/
  9. Reference # 94 is not in right format (e.g. year of publication is absent).
  10. Table 2 row “VISTA”: CA-170 is not targeting PD-L1 (column 3) see above (comment 12)
  11. “Another cytokine secreted by TFH cells is IL-10. High levels of IL-10 lead to a bad prognosis in AITL, probably through the higher levels of M2 macrophages that are polarized or because these high levels show an underlying hyperactivity of AITL” - Sentence is not clear please explain better.
  12. “The cytokines secreted by AITL also have a role in chemoattraction and activation of immune cells infiltrating AITL. For B cells, the upregulation of the CXCL13/CXCR5 axis recruits B cells in the AITL microenvironment (30,125). Other cells known to infiltrate AITL are CD8-positive T cells and Th17 cells, the latter being recurrently implicated in autoimmune events and being stimulated by IL-6 produced by mast cells in the microenvironment” in this point author just enumerate cytokines expressed by lymphoma cells and prosess of recruitment of different cells to lymphoma place but do not make conclusions what does this events mean (how they influence on lymphoma development).
  13. Figure 2 is seems like development of figure 1 and have the same focus. It is better to summarize them in one figure or make clear different focus for both figures (if it is different).
  14. In brief, the review looks unfinished and requires a major revision. it need  some essential work to be done. 1) in the end of each chapter short summarisation of the message are required; 2) checking that citation is present when authors make a statement in the text; 3) verification format of the literature are required;4)  Please, carefully check the gramma and style of the text; 5) work on the figures in pictures, and pictures subtitles are essential.
  15. In the review practically are absent conclusions in the end of the text.
  16. Stories about boxers does not increase review quality but rather mislead readers.
  17. Abstract should reflect the focus of the review and the main conclusions of review.

Minor comments

  1. Table 2 row “CTLA4”: comma have to be deleted in column 3.
  2. “By internalizing the exosomes in the immune system, the target cell phenotype is altered to pro-tumorigenic and pre-metastatic” Might be authors mean: “By interfering the exosomes with the immune system….
  3.  Authors use abbreviation “EVs” but do not explain it.

This is a very interesting topic and I hope that with a bit more work you can publish this paper soon!

Author Response

Reviewer 3:

Comments and Suggestions for Authors

The review focused on discussion microenvironment of lymphomas and interaction of different component of niche and immune cells during lymphoma formation and homeostasis.  The review will be interesting for scientists from a broad field of tumour research including, haematologists, immunologists, cellular biologists, vascular biologists and clinical doctors. The manuscript is well written and well-structured It has a good flow of text. This review presents some interesting data but there are obvious limitations and I suggest some points, which could be considered by the authors for manuscript revision.

 Thank you for the appreciation of our manuscript. We made all the required changes, in red, in the revised manuscript.

Major comments.

“This hypothesis was proven by the response to antibiotic treatment, which does not just lead to the eradication of the infection but also to a cease of tumoral development” This strong statement is not finish by paper citation or comment like “it is author opinion based on their data”.

 Thank you for your feed-back. We changed the phrase, which now is: “According to the Dutch group, based on their experience, this hypothesis was proven by the response to antibiotic treatment, which does not just lead to the eradication of the infection but also to a cease of tumoral development. “

“In contrary to B-cell lymphomas and Hodgkin’s lymphomas (HL), TCL show an increased heterogeneity regarding the TME”. Not clear in which regards TCL heterogeneity is increased in comparison with TME

Thank you for your feed-back. This sentence was indeed very confusing, and we decided to delete it.

Piece of the text below is not clear and have to be explanted by separation on “logical blocks”. “Independent from the subtype of TCL, the TME is considered as an entity of stimulation, sustenance, and promotion of lymphomas. Thus, by having a heterogenous composition, is a complex system that is dependent on the stimulation from tumor cells (13). The whole TME is dependent on the cell signals received by the tumour cells, which are also able to change their function because of genetic mutations”.

Thank you for your feed-back. We have re-written this paragraph, which now is “The TME is of crucial importance in lymphoma development, as it is constantly sustained by its heterogenous composition and stimulated by malignant cells (13). Thus, the TME is dependent on inter-cellular signalling between tumoral cells and surrounding cells, capable of changing their phenotype as a result of lymphoma cells acquiring new genetic mutations. “

Sentence “These are cytotoxic T lymphocytes (CTLs), T follicular helper cells (TFH), regulatory T cells (Tregs), natural killer (NK) cells and bystander B-Lymphocytes, as shown in Figure 1.” Author miss until this point the important component of TME. e.g tumor associated macrophages (TAMs) stromal components like fibroblasts, follicular dendritic cells, dendritic cells.

Thank you for your feed-back. The reviewer is correct. We have re-written the paragraph, which now is: “The TME is formed by a multitude of cells, which include tumor-associated macrophages, tumor-associated fibroblasts, follicular dendritic cells or dendritic cells, as well as immune cells, as is the case of cytotoxic T lymphocytes (CTLs), T follicular helper cells (TFH), regulatory T cells (Tregs), natural killer (NK) cells and bystander B-Lymphocytes (Figure 1).  “

On Figure 1 subtitle not described well. Most of component of the picture not indicated (e.g. vessels, M2-macrophages, Th17 lymphocytes, CD8+ -> CTL etc.) It is not clear all detail of the Figure. What big round cells mean. I guess this description of lymphoid tissue (like lymph node). it is not described in subtitles or text of the manuscript.

Thank you for your feed-back. We have re-written the Figure legend, which now is: “ Figure 1. The role of extracellular matrix in lymphoma development and progression, an exemplification of the sequence of event happening inside the malignant lymphoid tissue. The vascular network is disorganized, and it has larger pores compared to normal blood vessels, due to on over exposure to VEGF, PDGF-β, TGF-β. Due to these characteristics, the local microenvironment is hypoxic and there is a general lack of systemic immune cells provision, while the local immune cells secrete cytokines meant to facilitate lymphoma cell proliferation and invasion. The lymphoma cells (represented in the current figure through round brown), are at a higher number than all the other non-malignant immune cells in the lymphoma microenvironment. They arrive in the lymphoid organ through extravasation from general blood circulation, helped by locally modulated immunity, the lymphoma cells proliferate and become more aggressive, until they start to leave the lymphoid and enter into the lymphatic circulation. Inside the malignant lymphoid tissue, the macrophages are polarized to M2 phenotype and secrete Il-10, meaning that they have anti-inflammatory and malignant promoting properties. The TFH cells secrete IL-21, the TH17 and CD8+ T cells secrete IL-6.  TH17 cells have dual role by inhibiting Tregs, but also stimulating malignant proliferation of B cells. CD8+ cells (cytotoxic T cells-CTL) inhibit lymphoma progression, but they are in a small number in the lymphoma microenvironment.  However, the main mechanism through which the lymphoma progression is sustained by the extracellular environment is due to the secretion of exosomes by the immune cells (T or B lymphocytes) and the lymphoma cells (of B or T origin). In the top left part of the figure, in the circle, there is a representation of an exosome.“

“Recent data shows that in some lymphoma subtypes, cancer cells act similar to Tregs”. Similar to what? Please explain and supply reference.

Thank you for your feed-back. This sentence was indeed very confusing, and we decided to delete it.

Phrase is not clear «…. TME is most extensively studied in non-Hodgkin’s lymphomas (NHL), where the tumour cells mostly present as reduced from the tumoral mass».

Thank you for your feed-back. This sentence was indeed very confusing, and we decided to delete it.

Statement: “On the other side, in follicular lymphomas a strong correlation between the survival rate and the number of intrafollicular Tregs (FoxP3-positive) was reported with a high number of intrafollicular Tregs correlated with an unfavourable prognosis and survival.” has no reference present.

Thank you for your feed-back. We have re-written the sentence and added a reference: “On the other side, in follicular lymphomas a strong correlation between the survival rate and the number of intrafollicular Tregs (FoxP3-positive) was reported with a high number of intrafollicular Tregs correlated with an unfavourable prognosis and survival, as according to Xie et al. (16)“.

We added reference number 16:

  1. Xie M, Jiang Q, Zhao S, Zhao J, Ye X, Qian Q. Prognostic value of tissue-infiltrating immune cells in tumor microenvironment of follicular lymphoma: A meta-analysis. Int Immunopharmacol. (2020) 85:106684.

Authors use the term CD4 + as a synonym for T-regulatory cells, while the authors describe also T-helper cells that have the same phenotype (CD4+). Authors should be more precise in determining the phenotype of cells. Alternatively, they can remove CD4 as an exclusive definition of T-regulatory cells.

Thank you for your feed-back. We have removed the term CD4+ cells. When referring to regulatory T cells we used this term or Tregs. The same, when referring to T-helper cells, this is stated in the manuscript, to avoid any future confusions.

Authors describe “The T helper cells of type 1 (Th1) activate specifically the cytotoxic T cells (CD8+) through antigen presentation..” however they not describe what Th2 cells role in immunity and directly mention only “The Th2 cells on the other hand disrupt the activity of Th1” it is better, at least shortly, describe role of Th2.

Thank you for your feed-back. We have added in the revised manuscript that “T-helper 2 cells (Th2) are differentiated T lymphocytes important in the immune response against pathogens that do not directly infect cells. These cells also play key roles in tissue repair and contribute to pathophysiology of allergic disorders. “

Authors cited that “The blockage of VISTA and PD-1 through the use of small molecule, called CA-170, increases the T cell activation and IFN-γ production by T cells” however it was recently shown that CA-170 has no interaction with  PD-1/PD-L1 binding https://pubmed.ncbi.nlm.nih.gov/31374878/

Thank you for your feed-back. We have added in the revised manuscript that “Still, in Krakow, Musielak et al have shown that CA-170 has no interaction with PD-1/PD-L1 and state that this previously potential clinical application must be further investigated (95). “

We have also added reference 95:

  1. Musielak B, Kocik J, Skalniak L, Magiera-Mularz K, Sala D, Czub M, Stec M, Siedlar M, Holak TA, Plewka J. CA-170 - A Potent Small-Molecule PD-L1 Inhibitor or Not? Molecules(2019) 24(15):2804.

Reference # 94 is not in right format (e.g. year of publication is absent).

Thank you for your feed-back. We have relaced the initial cited manuscript with a more recent one, with more accurate information. The new reference is:

  1. Tagliamento M, Bironzo P, Novello S. New emerging targets in cancer immunotherapy: the role of VISTA. ESMO Open(2020) 4(Suppl 3):e000683.

Table 2 row “VISTA”: CA-170 is not targeting PD-L1 (column 3) see above (comment 12)

Thank you for your feed-back. We have corrected Table 2.

“Another cytokine secreted by TFH cells is IL-10. High levels of IL-10 lead to a bad prognosis in AITL, probably through the higher levels of M2 macrophages that are polarized or because these high levels show an underlying hyperactivity of AITL” - Sentence is not clear please explain better.

Thank you for your feed-back. We have corrected the phrase, which now is: “TFH cells also secrete IL-10, linked to a negative prognosis in AITL due to higher levels of M2 macrophages.“.

“The cytokines secreted by AITL also have a role in chemoattraction and activation of immune cells infiltrating AITL. For B cells, the upregulation of the CXCL13/CXCR5 axis recruits B cells in the AITL microenvironment (30,125). Other cells known to infiltrate AITL are CD8-positive T cells and Th17 cells, the latter being recurrently implicated in autoimmune events and being stimulated by IL-6 produced by mast cells in the microenvironment” in this point author just enumerate cytokines expressed by lymphoma cells and prosess of recruitment of different cells to lymphoma place but do not make conclusions what does this events mean (how they influence on lymphoma development).

Thank you for your feed-back. We have added that “By directly synthesizing IL-6, mas cells contribute to setting a pro-inflammatory TME. The AITL lymphoma clone itself attracts mast cells in this inflammatory microenvironment and thus mast cells change the immunological TME of AITL, with important clinical implications in diagnosis and prognosis. “.

Figure 2 is seems like development of figure 1 and have the same focus. It is better to summarize them in one figure or make clear different focus for both figures (if it is different).

Thank you for your feed-back. Indeed, the tow figures are very similar. We are more than happy to delete Figure 2 and only keep Figure 1, should the reviewer or editor decide so.

In brief, the review looks unfinished and requires a major revision. it need  some essential work to be done. 1) in the end of each chapter short summarisation of the message are required; 2) checking that citation is present when authors make a statement in the text; 3) verification format of the literature are required;4)  Please, carefully check the gramma and style of the text; 5) work on the figures in pictures, and pictures subtitles are essential.

Thank you for your feed-back. We have revised the manuscript and addressed all the above mentioned very constructive critiques. We now feel that the manuscript is much better, suitable for potential publication. Should any other suggestions be addressed, we are more than happy to work on the manuscript and bring it to a perfect, suitable for publication form.

In the review practically are absent conclusions in the end of the text.
Stories about boxers does not increase review quality but rather mislead readers.
Abstract should reflect the focus of the review and the main conclusions of review.

Thank you for the appreciation of our manuscript. Still, we would very much rather to keep the analogies with the boxing event, as it illustrates very good the interplay between B and T cells. The manuscript thus has a new, catchy title, introduction, and conclusion, that will be read and cited by physicians and scientists. We deeply appreciate your understanding and support.

We have also added in the Conclusion sections: “The tumor microenvironment and the local immune system of the niche have been merely regarded as the background for neoplastic cells to become a focus of our understanding of cancer development, cancer progression and point of action for new therapeutic concepts. The plasticity of the microenvironment makes it difficult to be studied in static circumstances and, for obvious reasons, in animal models as the constitution of the microenvironment and its interactions might differ from that in humans. Still, progress was been achieved for the contribution of immune system to lymphoma development and progression, and this knowledge has been transferred into therapeutic strategies. Future research will bring forward new insights with regard to inter- and intracellular signaling, metabolomics of the immune microenvironment, the impact of therapy and its predictive implications. “

We have also added in the abstract, in red, in the revised manuscript that “In this review, we aim to explain how T lymphocyte-driven control is linked to tumor development and describe the tumor-suppressive components of the resistant framework. This manuscript brings forward a new insight with regard to inter- and intracellular signaling, the immune microenvironment, the impact of therapy and its predictive implications. “

Minor comments

Table 2 row “CTLA4”: comma have to be deleted in column 3.

Thank you for your feed-back. We have corrected this mistake.

“By internalizing the exosomes in the immune system, the target cell phenotype is altered to pro-tumorigenic and pre-metastatic” Might be authors mean: “By interfering the exosomes with the immune system….

Thank you for your feed-back. We have corrected this mistake and the sentence is now: “By interfering the exosomes with the immune system, the target cell phenotype is altered to pro-tumorigenic and pre-metastatic. “

 Authors use abbreviation “EVs” but do not explain it.

Thank you for your feed-back. We have corrected this mistake and the abbreviation is now in the revised manuscript: extracellular vesicles (EVs). 

This is a very interesting topic and I hope that with a bit more work you can publish this paper soon!

Thank you for the appreciation of our manuscript. We have really appreciated all the very constructive critiques, have addressed them and we truly feel that the manuscript is much better, potentially suitable for publication, should the editor decide to endorse it for publication.

Round 2

Reviewer 2 Report

Dear Authors, thank you for providing the revised manuscript. However, I still have few concerns.

1. In their cover letter the Authors claim that "The manuscript thus has a new, catchy title, introduction, and conclusion.."

In the version 1 of their manuscript the title of the manuscript is: B cells versus T cells in the tumor microenvironment of malignant lymphomas. Are the lymphocytes playing the roles of Muhammad Ali versus George Foreman in Zaire 1974?

Same remains in the version 2: B cells versus T cells in the tumor microenvironment of malignant lymphomas. Are the 3 lymphocytes playing the roles of Muhammad Ali versus George Foreman in Zaire 1974?

Why authors claim that they provided a new title? Please explain.

2. Background section is still hard to follow due to its narrative style.

3. The references #17 and #94 are not in the correct format, having the serial number written two times.

Author Response

Dear Authors, thank you for providing the revised manuscript. However, I still have few concerns.

  1. In their cover letter the Authors claim that "The manuscript thus has a new, catchy title, introduction, and conclusion.."

In the version 1 of their manuscript the title of the manuscript is: B cells versus T cells in the tumor microenvironment of malignant lymphomas. Are the lymphocytes playing the roles of Muhammad Ali versus George Foreman in Zaire 1974?

Same remains in the version 2: B cells versus T cells in the tumor microenvironment of malignant lymphomas. Are the 3 lymphocytes playing the roles of Muhammad Ali versus George Foreman in Zaire 1974?

Why authors claim that they provided a new title? Please explain.

  1. Background section is still hard to follow due to its narrative style.

Thank you for the review. The reviewer is indeed correct. Still, we would very much like to keep this original title. I hope the reviewer understands and supports our intention. Still, we have modified the first chapter, with the background and have deleted most of the irrelevant data about the boxing. We have also re-organized this chapter, which is now better and more suitable for publication. The changes are marked with red, in the revised manuscript.

  1. The references #17 and #94 are not in the correct format, having the serial number written two times.

Thank you for the review. We have corrected this mistakes.

Reviewer 3 Report

Dear Authors, I have few additional comments to your last revision.

Major Comments

Line 123 “According to the Dutch group, based on their experience, this hypothesis was proven by the response…” Not clear which one Dutch group is mentioned in the manuscript. (University? lab? name clinic; Conference report which conference?; personal communication, with whom? Etc)

Minor comments

Line 158 “start to leave the lymphoid….”  Lymphoid tissue? Lymphoid organ?

Line 166 “In the top left part of the figure…” Do Authors mean “in the top right”?

Line 419 “By directly synthesizing IL-6, mas cells…” Do you mean “Mast cells”?

Line 450  it better to change the phrase like it is more clear “By exosomes interfering with the immune system, the target cell phenotype is altered to pro-tumorigenic (148) and pre-metastatic”

Alternatively, you could use your previous variant but with clear indication where exosomes have internalized e.g. “By the exosomes internalizing in cells of immune system, the target cell phenotype is altered to pro-tumorigenic (148) and pre-metastatic”

Please check you text again. 

Best wishes 

Author Response

Reviewer 3:

Major Comments

Line 123 “According to the Dutch group, based on their experience, this hypothesis was proven by the response…” Not clear which one Dutch group is mentioned in the manuscript. (University? lab? name clinic; Conference report which conference?; personal communication, with whom? Etc)

Thank you for the review. We have corrected the sentence, which now is : “According to the van den Berg group, from the University of Groningen, based on their experience, this hypothesis was proven by the response to antibiotic treatment, which does not just lead to the eradication of the infection but also to a cease of tumoral development. “

Minor comments

Line 158 “start to leave the lymphoid….”  Lymphoid tissue? Lymphoid organ?

Thank you for the review. We have corrected this mistake.

Line 166 “In the top left part of the figure…” Do Authors mean “in the top right”?

Thank you for the review. We have corrected this mistake.

Line 419 “By directly synthesizing IL-6, mas cells…” Do you mean “Mast cells”?

Thank you for the review. We have corrected this mistake.

Line 450  it better to change the phrase like it is more clear “By exosomes interfering with the immune system, the target cell phenotype is altered to pro-tumorigenic (148) and pre-metastatic”

Thank you for the review. We have corrected this mistake.

Alternatively, you could use your previous variant but with clear indication where exosomes have internalized e.g. “By the exosomes internalizing in cells of immune system, the target cell phenotype is altered to pro-tumorigenic (148) and pre-metastatic”

Thank you for the review. The first version of the sentence is perfect. Thank you

Please check you text again.

Thank you for the review, as well for the excellent advices to improve our manuscript.

Best wishes

Thank you greatly.

Round 3

Reviewer 2 Report

Dear Authors,

Thank you for providing the second revision of your manuscript. I have no further comments.

Author Response

Reviewer 2:

Comments and Suggestions for Authors
Dear Authors,

Thank you for providing the second revision of your manuscript. I have no further comments.

Thank you greatly for your support.
